# Characterization of the *Burkholderia cenocepacia* J2315 Surface-Exposed Immunoproteome

**DOI:** 10.3390/vaccines8030509

**Published:** 2020-09-06

**Authors:** Sílvia A. Sousa, António M.M. Seixas, Manoj Mandal, Manuel J. Rodríguez-Ortega, Jorge H. Leitão

**Affiliations:** 1iBB–Institute for Bioengineering and Biosciences, 1049-001 Lisbon, Portugal; antonio.seixas@tecnico.ulisboa.pt (A.M.M.S.); manoj2biomedical@gmail.com (M.M.); 2Departament of Biochemistry and Molecular Biology, Córdoba University, 14071 Córdoba, Spain; mjrodriguez@uco.es

**Keywords:** *Burkholderia cepacia* complex, surface-exposed moieties, liquid chromatography–tandem mass spectrometry (LC-MS/MS), immunoreactive proteins, cystic fibrosis, serum samples, immunoreactivity

## Abstract

Infections by the *Burkholderia cepacia* complex (Bcc) remain seriously life threatening to cystic fibrosis (CF) patients, and no effective eradication is available. A vaccine to protect patients against Bcc infections is a highly attractive therapeutic option, but none is available. A strategy combining the bioinformatics identification of putative surface-exposed proteins with an experimental approach encompassing the “shaving” of surface-exposed proteins with trypsin followed by peptide identification by liquid chromatography and mass spectrometry is here reported. The methodology allowed the bioinformatics identification of 263 potentially surface-exposed proteins, 16 of them also experimentally identified by the “shaving” approach. Of the proteins identified, 143 have a high probability of containing B-cell epitopes that are surface-exposed. The immunogenicity of three of these proteins was demonstrated using serum samples from Bcc-infected CF patients and Western blotting, validating the usefulness of this methodology in identifying potentially immunogenic surface-exposed proteins that might be used for the development of Bcc-protective vaccines.

## 1. Introduction

The *Burkholderia cepacia* complex (Bcc) presently comprises at least 23 closely related bacterial species [1,2,3] that emerged in the 1980s as important pathogens for patients suffering from cystic fibrosis (CF). Bcc bacteria can be found in a wide range of environments including water, soil and the rhizosphere of plants, being able to survive in hospital settings and pharmaceutical aqueous solutions [4]. Despite their presence in the environment, Bcc bacteria are opportunistic pathogens capable of causing life-threatening infections in the respiratory tract of immunocompromised patients, patients with chronic granulomatous disease (CGD) and especially patients suffering from CF, the most common lethal inherited genetic disease among Caucasians [5]. The majority of Bcc-infected CF patients experience an increased decline in pulmonary function, which is associated with chronic infection and exacerbation episodes [5]. However, some patients can also develop a rapid and fatal necrotizing pneumonia known as the cepacia syndrome [5,6]. These factors, coupled with the organism’s armory of virulence factors and the intrinsic resistance of Bcc bacteria to several clinically available antimicrobials, render chronic infections virtually untreatable [5,7,8,9], making infections by Bcc a major concern since the clinical outcome is highly variable and unpredictable [10]. Considering all the above, a strategy to eradicate Bcc bacteria from CF patients is imperative. Since there is no eradication therapy presently available [11], new strategies to deal with Bcc infections are of major interest. Currently, no such strategies exist, but a wide variety of approaches and antigens are under study for the development of a possible vaccine against Bcc [12,13,14,15]_._

Bacteria use the proteins expressed on their surface to interact with their environment, being also exposed to the immune system of the host. Therefore, surface-exposed proteins are appealing antigens for the development of new therapeutic strategies for the eradication of bacterial infections [16]. However, a wide range of characteristics including low abundance, hydrophobic natures and low solubility make these proteins difficult to study, especially using first-generation gel-dependent proteomics approaches [17,18,19]. In order to uncover novel surface-exposed antigens to be studied for the development of new therapeutic strategies for the eradication of Bcc infections, a bioinformatics analysis of the surface-exposed proteins of *B. cenocepacia* J2315 and their immune profile was performed in this work. The proteins identified were experimentally validated using a surface “shaving” approach comprising the digestion of live intact cells with proteases and the recovery of the cells’ surface-exposed moieties, followed by their analysis by liquid chromatography–tandem mass spectrometry (LC-MS/MS) (Figure 1). In this way, using a sensitive and gel-free approach, the limitations associated with membrane protein analysis can be overcome. This approach also grants the acquisition of new information, as it allows the identification of the domains of the identified proteins that are more accessible to proteases and therefore to the host immune system and antibodies [17]. The surface “shaving” approach has been applied mainly to Gram-positive bacteria, owing to their characteristic thicker cell walls that confer a greater resistance to cell lysis. Nonetheless, it has also been applied to Gram-negative bacteria, unicellular fungi and also larvae of parasitic helminth worms [17,20,21,22]. Different levels of success were reported in each case, depending on factors such as protein structure and microorganism surface topology.

Despite the increase in popularity of this technique, it had never been applied to bacteria of the Bcc, and therefore, this pioneering work was envisioned. With this purpose, and to discover surface proteins exposed to the host during the infection process, the protocol was optimized for *B. cenocepacia* J2315 using growth conditions that mimic the environment in the lungs of CF patients. The CF lung has steep oxygen gradients, usually with low oxygen concentrations within the typically thick mucus layer [23]. Bcc bacteria can also survive intracellularly in host cells, where they are exposed to low pH, severe nutrient limitation and oxidative stress [24].

The bioinformatics approach used in this work allowed us to identify 263 surface-exposed proteins, 143 of them having a high probability of containing predicted immunogenic surface-exposed regions. Using the experimental surface-shaving optimized protocol, we were able to confirm that 16 of the identified proteins were in fact surface-exposed, and their exposed moieties were identified. Three selected proteins were cloned, overexpressed, purified and confirmed as immunoreactive with the sera of CF patients with a record of infections with Bcc.

## 2. Materials and Methods

### 2.1. Bacterial Strains and Culture Conditions

The bacterial strains and plasmids used are shown in Table 1. PIA (Pseudomonas Isolation Agar, Becton Dickinson, Heidelberg, Germany) and Lennox broth (LB, containing, in g/L, tryptone, 10; yeast extract, 5; NaCl, 5; and agar, 20) were used, respectively, to maintain *B. cenocepacia* J2315 and *Escherichia coli* strains. When appropriate, LB was supplemented with 150 μg/mL ampicillin or 50 μg/mL kanamycin. Unless otherwise stated, bacteria were cultivated in shaking flasks (250 rev/min) containing LB liquid at 37 °C, supplemented with adequate antibiotics.

### 2.2. Digestion of Live B. cenocepacia Cells’ Surface with Trypsin

The generation and recovery of peptides by “shaving” *B. cenocepacia* J2315 cells with trypsin was performed based on the work of Rodríguez-Ortega (2018) with slight adaptations [27]. After bacterial growth until mid-exponential phase under the conditions indicated in Materials and Methods, the cultures’ optical densities at 600 nm (OD_600_) were normalized to 0.5. Aliquots of 100 µL of the bacterial suspensions were spread onto the surfaces of Artificial Sputum Medium (ASM) agar plates. The composition of the ASM was formulated to mimic CF patients’ sputum. The ASM solid medium contained, in g/L, porcine stomach mucin (Sigma-Aldrich, St. Louis, MO, USA) 5.0; low molecular-weight salmon sperm DNA (Sigma-Aldrich, St. Louis, MO, USA), 4.0; NaCl (Sigma-Aldrich, St. Louis, MO, USA), 5.0; KCl (Sigma-Aldrich, St. Louis, MO, USA), 2.2; casamino acids (Difco Laboratories, Detroit, MI, USA), 5.0; Tris Base (Sigma-Aldrich, St. Louis, MO, USA), 1.81; and agar, 20 (pH 7.0), with 5.0 mL/L of egg yolk emulsion as a source of lecithin (Sigma-Aldrich, St. Louis, MO, USA) and 5.9 mg/L of the iron-chelating agent diethylene triamine pentaacetic acid (DTPA) (Sigma-Aldrich, St. Louis, MO, USA) [28]. After incubation for 22 h at 37 °C under aerobic conditions, cells were scraped from the ASM agar plates’ surface with phosphate-buffered saline solution (PBS; 8.18 g/L NaCl, 0.2g/L KCL, 2.68 g/L Na_2_HPO_4_ and 0.245 g/L NaH_2_PO_4.,_ pH 7.4) and harvested by centrifugation at 3500× *g* for 5 min at 20 °C. The pelleted bacteria were washed thrice with PBS and finally resuspended in 1 mL of PBS. Tryptic digestion was carried out with 2.5 μg/mL porcine trypsin (sequencing grade; Promega, Madison, WI, USA) for 10 min at 37 °C with gentle agitation. The digestion mixtures were centrifuged at 3500× g for 5 min at 4 °C, and the supernatants (the “surfome” containing the peptides) were filtered using 0.22 μm-pore-sized syringe filters (Whatman^TM^, Buckinghamshire, UK). The surfome was re-digested with 1 μg/mL trypsin overnight at 37 °C with gentle agitation. The solution containing the peptides was stored at −80 °C until further analysis. Aliquots (15 µL) of the tryptic digestion mixtures were separated by 15% SDS–polyacrylamide gel electrophoresis (PAGE). The gel was silver stained according to the protocol of Chevallet et al. (2006) [29].

### 2.3. Bacterial Cell Viability Assay

For the analysis of bacterial viability, 10 μL of the bacterial suspensions were taken pre- and post-trypsin digestion. The samples were serially diluted (10^−1^ to 10^−8^) with 0.9% (W/V) NaCl, and the different dilutions were spotted on LB agar plates. The plates were incubated overnight at 37 °C, and the total colony forming units (CFUs) were determined for each sample.

### 2.4. Cleaning of Peptides with Solid-Phase Extraction Cartridges

After bacterial cell removal, the peptides that resulted from the digestion of the live bacterial cells with trypsin were cleaned and concentrated using Oasis HLB extraction cartridges (Waters, NY, USA), following the manufacturer’s instructions and as modified by Rodríguez-Ortega [27]. Briefly, 150 µL samples were loaded into extraction cartridges previously conditioned with 80% acetonitrile (ACN), followed by 0.1% formic acid solution. Peptides were eluted from the extraction cartridges using increasing concentrations of ACN (10, 20 and 50%) in 0.1% formic acid. Peptide fractions were dried using a vacuum concentrator (Eppendorf, Hamburg, Germany), resuspended in 100 µL of a solution containing 2% ACN and 0.1% formic acid and kept at −20 °C until further processing.

### 2.5. MALDI-TOF MS Analysis

The efficiency of the tryptic digestion of the live bacterial cells was checked at the qualitative level by looking at the mass spectrum from MALDI-TOF MS. Aliquots of 1 μL of the cleaned peptides were mixed with 1 μL of matrix solution (α-cyanohydroxycinnamic acid at a concentration of 5 mg/mL in 70% ACN/ 0.1% trifluoroacetic acid) and spotted onto a MALDI plate using the dry-droplet method. A 4800 Proteomics Analyzer MALDI-TOF/TOF Mass Spectrometer (Applied Biosystems, Waltham, MA, USA) was used to acquire the mass spectra, in the *m/z* range of 800 to 4000, with an accelerating voltage of 20 kV in reflectron mode. The spectra were internally calibrated using peptides from trypsin autolysis ([M + H^+^] = 842.509, [M + H^+^] = 2211.104) with an *m/z* precision of ± 20 ppm.

### 2.6. LC-MS/MS Analysis

A Dionex Ultimate 3000 nano UPLC (Thermo Scientific, San Jose, CA, USA), equipped with a reverse phase C18 75 μm × 50 mm Acclaim PepMap Column (Thermo Scientific) was used for peptide separation. The system was operated for a total run time of 85 min at a flow rate of 300 nL/min and 40 °C. The peptide mix was previously concentrated and cleaned up using a 300 μm × 5 mm Acclaim PepMap cartridge (Thermo Scientific) in 2% ACN/0.05% formic acid for 5 min, at a 5 µL/min flow rate. The mobile phase used for the chromatographic separation was composed of Solution A (0.1% formic acid) and Solution B (80% ACN, 0.1% formic acid). The elution conditions used were as follows: 4–35% Solution B for 60 min, 35–55% Solution B for 3 min, and 55–90% Solution B for 3 min, followed by 8 min of washing with 90% Solution B and re-equilibration for 12 min with 4% Solution B. Peptide positive ions were eluted and ionized by a nano-electrospray ionization source and further analyzed in positive mode on a trihybrid Thermo Orbitrap Fusion (Thermo Scientific) mass spectrometer, operating in the Top30 Data-Dependent Acquisition mode. A 3 s maximum cycle time was used. Peptide-precursor single MS scans were acquired in a 400–1500 *m/z* range at 120,000 resolution (at 200 *m/z*), with a 4 × 10^5^ ion count target threshold. For MS/MS, precursor ions were previously isolated in the quadrupole at 1.2 Da and then collision induced fragmentation (CID)-fragmented in the ion trap with 35% normalized collision energy. The monoisotopic precursor selection was turned on. The ion trap parameters were (i) the automatic gain control was 2 × 10^3^, (ii) the maximum injection time was 300 ms, and (iii) only those precursors with charge state 2–5 were sampled for MS/MS. In order to avoid redundant fragmentations, the dynamic exclusion time was set to 15 s, with a 10 ppm tolerance around the selected precursor and its isotopes.

### 2.7. Protein Identification by Database Searching

The mass spectrometry raw data were processed using the Proteome Discoverer software (version 2.1.0.81, Thermo Scientific). Charge state deconvolution and deisotoping were not performed. The SEQUEST engine was used to search the MS/MS spectra against the *B. cenocepacia* J2315 genome database downloaded from UniProt (www.uniprot.org), using the following search parameters: one missed cleavage allowed for trypsin digestion, methionine oxidation set as a variable modification, a value of 10 ppm set for the mass tolerance for the precursor ions, and a 0.1 Da tolerance for the ion products. Peptide identifications were accepted only when they exceeded the filter parameter Xcorr *score* versus charge state with SequestNode Probability Score (+1 = 1.5, +2 = 2.0, +3 = 2.25, +4 = 2.5). Peptide spectral matches (PSM) were validated at a 1% FDR, using a percolator based on q-values. *B. cenocepacia* J2315 surface-associated proteins identified using the “surface shaving” were characterized with regard to subcellular localization and associated biological and molecular functions using The *Burkholderia* Genome Database, PSORTb 3.0.2 and Pfam 33.1 bioinformatics tools [30,31,32]. The mass spectrometry raw data have been deposited at PeptideAtlas (www.peptideatlas.org), with the dataset identifier PASS01615.

### 2.8. Molecular Biology Techniques

Total DNA was extracted from *B. cenocepacia* strain J2315 cells harvested from liquid cultures at the exponential growth phase using the High Pure PCR Template Preparation Kit (Roche, Basel, Switzerland). Techniques including plasmid isolation and purification (NZYTech, Lisbon, Portugal), DNA amplification, DNA restriction and T4 DNA ligation (Thermo Fisher Scientific), agarose gel electrophoresis, SDS-PAGE, and *E. coli* transformation were performed using standard procedures [33]. The primers used for the amplification of the gene *BCAL2645* were UP-BCAL2645 (5’-TGACATATGAACATGAAAATCGC-3’) and LW-BCAL2645 (5’-AACTCGAGCTGATGCTGTTGC-3’), containing the NdeI and XhoI restriction sites (underlined), respectively, at their 5’ ends. Nested polymerase chain reaction (PCR) was used to amplify the gene *BCAL2022*. This involved two sequential amplification reactions. In the first reaction, the primer pair used was Nested_UP-BCAL2022 (5′-TTTCAACCACGGAGGATTTC-3′) and Nested_LW-BCAL2022 (5′-GACAGCAACATCAGCGAGAG-3′). The second PCR reaction used the product of the first PCR reaction as the template and the primers UP-BCAL2022 (5′-TTCATATGTCGCTTTTCGACTC-3′) and LW-BCAL2022 (5′-AAGGTACCCTGCGCGGGCG-3′), containing, respectively, the NdeI and KpnI restriction sites (underlined) at their 5′ ends. All the primers were designed based on the *B. cenocepacia* J2315 genome sequence (available at [34]) and acquired from STAB VIDA (Portugal) via a paid service.

### 2.9. Cloning and Overexpression of B. cenocepacia J2315 bcal2022, bcal2645 and bcal2958 Proteins

The gene BCAL2645 was cloned using the plasmid pET23a+ and the 659 bp PCR product obtained using the primers UP-BCAL2645 and LW-BCAL2645, digested with the restriction enzymes NdeI and XhoI. The BCAL2645 fragment was ligated into the NdeI/XhoI-digested pET23a+, yielding pSAS36. The gene BCAL2022 was cloned in the plasmid pSAS38. The 681 bp PCR product obtained using primers UP-BCAL2022 and LW-BCAL2022 was digested with the restriction enzymes NdeI and KpnI, and the BCAL2022 fragment was ligated into the NdeI/KpnI-digested pSAS38, yielding the construct pMM1. The plasmid pSAS38 was previously created by cloning the thrombin recognition site of pET29a^+^ in the pET23a^+^. The nucleotide sequences of the cloned fragments were confirmed by sequencing (Eurofins Genomics, Ebersberg, Germany).

Plasmids pSAS36 and pMM1 allow, respectively, the controlled expression of C-terminus 6× His-tag derivatives of the proteins BCAL2645 and BCAL2022, upon induction of the T7 promoter by isopropyl β-D-thiogalactoside (IPTG). *E. coli* BL21 (DE3) transformed with each plasmid were cultivated in 100 mL of LB liquid medium supplemented with 150 μg/mL ampicillin at 37 °C (pSAS36) or 30 °C (pMM1) and with orbital shaking (250 rpm). When the cultures reached an OD_640_ of 0.6, IPTG was added to a 0.4 mM final concentration and the cultures were incubated for an additional 2 h under the same temperature and orbital-agitation conditions. Bacteria were harvested by centrifugation for 5 min at 7000× *g* and 4 °C. The resulting pellets were resuspended in 10 mL of sonication buffer (20 mM sodium phosphate, 500 mM NaCl, 20 mM imidazole, pH 7.4) and stored at −80 °C until further processing. Recombinant 6× His-tagged BCAL2645 and BCAL2022 production was assessed by SDS-PAGE analysis, followed by immunoblotting experiments using a monoclonal anti-polyhistidine peroxidase conjugate clone HIS-1 antibody (diluted 1:2000, Sigma, St. Louis, MO, USA), as previously described [26]. BCAL2958 overexpression was performed as previously described [26].

### 2.10. Purification of B. cenocepacia J2315 His-Tagged Proteins BCAL2022, BCAL2645 and BCAL2958

Bacterial suspensions of recombinant *E. coli* used to overexpress proteins were lysed by ultrasonic vibration with a Branson sonifier 250 (Branson Ultrasonics, Brookfield, CT, USA), using 6 cycles of sonication of 30 s each with a 40% duty cycle, adding 2% (v/v) Triton X-100 before the last two sonication cycles. Centrifugation at 12,000× *g* for 30 min at 4 °C was then performed to remove non-soluble cell debris. The cleared supernatants were collected and kept at 4 °C. The BCAL2645 and BCAL2022 6× His-tagged proteins were then purified by affinity chromatography using HisTrap FF columns (GE Healthcare, Chicago, IL, USA). The columns were initially equilibrated by flowing 10 mL of Buffer A (20 mM sodium phosphate, 750 mM NaCl, 20 mM imidazole, 10% glycerol, pH 7.4) for BCAL2645 or Buffer B (20 mM sodium phosphate, 500 mM NaCl, 20 mM imidazole, pH 7.4) for BCAL2022. The proteins were then eluted with 5 mL of Buffer A or B, respectively, containing imidazole concentrations of 60, 100, 150, 200, 250, 300, 400 and 500 mM. Aliquots of 1 mL were collected for each protein, followed by their analysis by SDS-PAGE. Immunoblotting experiments were carried out as previously described using a commercial monoclonal anti-polyhistidine peroxidase conjugate clone HIS-1 antibody (Sigma; St. Louis, MO, USA) [26]. BCAL2958 purification was performed as previously described [26].

### 2.11. CF Patients’ Blood Sera’s Immunoreactivity against the Proteins BCAL2022, BCAL2645 and BCAL2958

The His-tagged purified proteins BCAL2022, BCAL2645 and BCAL2958, as well as the negative control bovine serum albumin (BSA) (Nzytech), were loaded into 15% SDS-PAGE gels and subjected to electrophoresis for 1 h at 160 V using standard procedures [33]. The gels were then equilibrated for 15 min in transfer buffer (48 mM Tris, 39 mM glycine, 20 % (V/V) methanol, 0.04 % (W/V) SDS, pH 9.2). After this time, the proteins were electrotransferred to nitrocellulose (NC) membranes (PALL corporation, Port Washington, NY, USA) using a Trans-Blot^®^ SD (BIORAD, Hercules, CA, USA) device apparatus at 15 V and 120 mA for 1 h. After protein electrotransfer, the membranes were blocked overnight at 4 °C with 5 % (w/v) skim milk (Difco Laboratories, Detroit, MI, USA) in PBS. The membranes were then probed for 3 h at room temperature with individual serum samples from CF patients (1:1000 dilution) or with a commercial pool of human sera from healthy donors (1:1000 dilution, S7023, Sigma, St. Louis, MO, USA). Serum samples were obtained from blood samples collected from two CF patients with a clinical record of Bcc infection who attended the Santa Maria hospital in Lisbon, Portugal, as previously described [19]. The membranes were washed with PBS-T (PBS containing 0.05 % (v/v) Tween 20) and then incubated for 1 h at room temperature with a secondary antibody horseradish peroxidase (HRP)-conjugated rabbit anti-human IgG (1:5000 dilution, Santa Cruz Biotechnology, Dallas, TX, USA). After the removal of the secondary antibody and a wash with PBS-T, the membranes were treated with the peroxidase substrate ECL (Sigma, St. Louis, MO, USA) and signals were detected using the FUSION Solo apparatus (Vilber Lourmat, Collégien, France).

### 2.12. Enzyme-Linked Immunosorbent Assay (ELISA)

Quantification of the IgG levels present in the serum samples from CF patients with a clinical history of Bcc infection against purified 6× His-tagged BCAL2645 was performed by enzyme-linked immunosorbent assay (ELISA) as previously described [26]. Briefly, the BCAL2645 protein was diluted to 2 μg/mL in 100 mM sodium carbonate buffer (pH 9.6), and 100 μL was applied per well to 96-well ELISA plates (Greiner Microlon 600, Greiner Bio-One, Kremsmünster, Austria). After overnight incubation at 4 °C, the plates were blocked with 250 μL of 3% BSA/PBS overnight at 4 °C. The serum samples to be tested were serially diluted (1:100 to 1:10,000) in 3% BSA/PBS-T (PBS supplemented with 0.05% Tween 20). After the addition of the diluted serum samples to the plates, they were incubated 2 h at 25 °C. Then, the plates were washed with PBS-T, and 100 μL of HRP-conjugated rabbit anti-human IgG (Santa Cruz Biotechnology) antibody at 1:3000 dilution in 3% BSA/PBS-T was added to the plates. After incubation at 25 °C for 1 h, the plates were washed with PBS-T. Then, 100 μL of the peroxidase substrate 3,3′,5,5′-tetramethylbenzidine (TMB, Sigma-Aldrich, St. Louis, MO, USA) were added. After incubation for 20 min at 25 °C, the reaction was stopped by the addition of 100 μL of 0.5 M H_2_SO_4_. The plates were read at 450 nm in the SPECTROstar Nano Microplate Reader (BMG LABTECH, Ortenberg, Germany). A pool of sera from healthy humans (Sigma-Aldrich, St. Louis, MO, USA) was used as control. Internal positive and negative controls were included in each plate. All serum samples were analyzed in triplicate with at least two independent experiments.

Serum antibody concentrations were defined as endpoint titers and were calculated as the reciprocal of the highest serum dilution producing an OD_450_ above the cut-off value. The cut-off value was determined as the mean OD_450_ of the negative control plus 3 standard deviations. A titer above the cut-off value was considered a positive result for the ELISA.

### 2.13. Bioinformatics Analysis

The proteins identified by bioinformatics tools as extracellular or present in the outer membrane were selected using the Burkholderia Genome Database [30]. With this purpose, a search using the subcellular localization annotation was performed. The amino acid sequences of the proteins selected for being extracellular or present in the outer membrane were retrieved from the genome sequence of *B. cenocepacia* J2315, available at the above-referenced database. The sequences were further analyzed for B-cell epitopes using BepiPred-2.0, in the immune epitope database (IEDB), using a 0.5 threshold [35]. This 0.5 threshold was used because it is the point at which sensitivity and specificity are both maximized in BepiPred 2.0. Peptides shorter than five or larger than 25 amino acids were not considered, as B-cell epitopes are usually within this size range [35].

## 3. Results

### 3.1. Identification and Analysis of Surface-Exposed Proteins of B. cenocepacia J2315 with Immunogenic Epitopes Using Bioinformatics Tools

The Burkholderia Genome Database [30] was used to search for extracellular or outer membrane-encoding genes. A total of 161 genes putatively coding for outer membrane proteins and 102 genes coding for putative extracellular proteins were identified (Appendix A). The amino acid sequences of these proteins were then retrieved from the genome sequence of *B. cenocepacia* J2315, available at the above-referenced database. B-cells are a fundamental component of the adaptive immune system, having the ability to recognize and provide long-term protection against infectious pathogens [36]. These cells produce antibodies that recognize their molecular target (called antigen) by binding to a part of it, the epitope, in a highly selective manner. This recognition process can be used for the development of vaccines to provide long-term protection against infectious pathogens. Therefore, the putative 263 surface-exposed proteins identified were inspected for linear B-cell epitopes using the BepiPred-2.0 tool. Peptides shorter than five or larger than 25 amino acids were not considered, as B-cell epitopes are usually within this size range [35]. A total of 143 proteins were found as having a high probability of being immunogenic, presenting an average of B-cell epitopes higher than 0.5. The predicted B-cell epitopes for each protein identified are shown in Appendix A.

### 3.2. Characterization of Surface-Exposed Proteins Using Trypsin Digestion of Live B. cenocepacia J2315 Cells

In order to validate the bioinformatically identified proteins as surface-exposed proteins, a surface-shaving protocol using trypsin was optimized for live cells of *B. cenocepacia* J2315, grown in conditions that mimic the CF lung, combined with the subsequent LC-MS/MS analysis of the generated peptides.

#### 3.2.1. Optimization of “Shaving” Protocol

In order to mimic the CF lung environment, *B. cenocepacia* J2315 was inoculated in ASM agar plates. ASM is composed of components similar to the constituents of CF sputum, including free DNA, mucin, a lecithin source, an iron-chelating agent, salts and amino acids at concentrations similar to those in the average CF patient’s sputum [28]. The protocol for the surface “shaving” with trypsin of live cells is a compromise between the efficiency of protease-generated peptides and the avoidance of contamination with cytoplasmic proteins [37]. Therefore, to obtain minimal cell lysis of *B. cenocepacia* J2315, the protocol for the surface shaving with trypsin of live cells was optimized. Several buffers previously used for “shaving” protocols for other bacteria were tested, such as PBS containing 30% sucrose (pH 7.4), PBS containing 10% sucrose (pH 7.4), 10 mM HEPES (pH 7.4) and PBS (pH 7.4) [20]. Therefore, aliquots before and after the digestion of *B. cenocepacia* cells with trypsin (30 min at 37 °C) were serially diluted and spot inoculated on LB agar to determine total CFUs. The PBS buffers supplemented with sucrose and the HEPES buffer resulted in the lysis of more than 54% of the *B. cenocepacia* cells. Instead, a 22% lysis was determined for the cells digested with PBS. Therefore, PBS was selected, different concentrations of trypsin (2.5 or 5 μg/mL) and incubation times (10 or 30 min) at 37 °C were also tested, and the aliquots containing the tryptic digestion mixtures were separated by SDS-PAGE and visualized after silver staining (Figure 2). After the ”shaving” protocol optimization, the best condition that enabled lower *B. cenocepacia* J2315 cell lysis and higher digestion of the exposed domains of surface proteins with trypsin was the incubation of cells with 2.5 μg/mL in PBS for 10 min.

After digestion with trypsin, bacterial cells were removed by centrifugation and the supernatants were filtered to remove any remaining bacterial cells. To determine if one trypsin digestion was sufficient to generate small peptides for LC-MS/MS analysis, the surfome samples before and after re-digestion with 1 μg/mL trypsin overnight at 37 °C were analyzed by MALDI-TOF MS analysis (Figure 3). As expected, the second digestion led to a significant increase in the number of peptides. Therefore, a second digestion with trypsin was performed before LC-MS/MS analysis.

#### 3.2.2. Identification of Surface-Exposed Proteins Using Trypsin Digestion of Live *B. cenocepacia* J2315 Cells

The trypsin-generated peptides were analyzed by LC-MS/MS. Upon the validation of peptide spectral matches, it was possible to identify 1169 proteins. Considering that the *B. cenocepacia* J2315 genome harbors 7117 coding sequences (CDS), the surface-shaving method used retrieved 16.4% of the predicted CDSs. However, to avoid false-positive identification, we used a threshold score of ≥5 unique peptides within a single protein, according to the recommendations of Higdon and Kolker (2007) [38]. Using this criterion, a total of 333 proteins were selected for the further bioinformatics analysis of subcellular localization with PSORTb and function analysis with Pfam (Appendix A, Figure 4).

A total of 55 predicted surface-exposed proteins (13 outer membrane proteins, 3 extracellular and 39 unknown) were identified (Table 2). These proteins were further analyzed using BepiPred 2.0 to study the probability of the exposed moieties being immunogenic (Appendix A). A large fraction of the proteins identified had predicted intracellular subcellular localization (222 cytoplasmic proteins; 15 from the cytoplasmic membrane and 41 from the periplasm). However, around 63% of orthologues of these identified proteins (164 cytoplasmic proteins; seven from the cytoplasmic membrane and four from the periplasm) were previously reported to be intracellular/surface moonlighting proteins in other Gram-negative bacteria [39]. 

### 3.3. Characterization of B. cenocepacia bcal2022 and bcal2645 Proteins’ Immunoreactivity

In order to validate the results from the LC-MS/MS identification of surface-exposed proteins and bioinformatics analysis of putative immunogenic proteins, we selected three proteins identified by the surfomics approach and with a high probability of comprising surface-exposed immunogenic peptides by bioinformatics for further analysis. These include the outer membrane protein A (OmpA)-like proteins BCAL2958 and BCAL2645, and BCAL2022, a phage-shock protein A (PspA)-like protein (Table 3).

In order to produce the proteins BCAL2645 and BCAL2022, the *BCAL2645* and *BCAL2022* genes were cloned into the expression vector pET23a+ under the control of the T7 promoter. Then, the overexpression of the proteins was achieved in *E. coli* as 6× His-tagged derivatives by induction with IPTG. The recombinant proteins were then purified to homogeneity by nickel affinity chromatography (Figure 5A). To test the immunoreactivity of the proteins BCAL2645 and BCAL2022, the purified fraction of each of the proteins was used for the detection of IgG antibodies against each protein in two serum samples from CF patients with a clinical record of Bcc infection (Figure 5A). BCAL2958 was used as positive control, and bovine serum albumin fraction V (BSA) was used as a negative control. Similarly to BCAL2958, the BCAL2465 protein exhibited a strong immunoreactivity against the tested sera from the CF patients infected with Bcc, indicating that the immunological systems of the CF patients were exposed to this protein during the infection and developed antibodies against it, therefore showing it to be a good candidate for immunotherapies. On the other hand, the BCAL2022 protein was only immunoreactive with one of the serum samples tested (Figure 5A) and therefore was not used in ELISA experiments. These results suggest that although the gene encoding the BCAL2022 protein is highly conserved in Bcc [19], the expression and surface-exposure of the protein is not a common trait of the Bcc bacteria during the infection of CF patients. No reactivity of the tested proteins was observed when using samples of a pool of sera from healthy individuals (Figure 5B).

Analysis of the BCAL2958, BCAL2645 and BCAL2022 amino acid sequences using the ProtParam program revealed that these proteins have predicted molecular weights of 23.9, 21.6 and 24.3 kDa, respectively. After SDS-PAGE, at least three different-molecular-weight forms of the protein BCAL2958 and two such of BCAL2645 were apparent (Figure 5A). The multiple bands most probably resulted from the different stages of processing and translocation of the proteins from the cytoplasm to the outer membrane, as previously reported for the BCAL2958 protein and for other OmpA-like proteins [26]. In the case of BCAL2958, the first two forms with higher molecular masses were described to correspond to the native pre-protein with a 6× His-tag containing the signal peptide, and the mature protein with a 6× His-tag without the signal peptide, respectively [26]. The serum samples from CF patients with Bcc infection records presented high antibody titers against BCAL2645 and BCAL2958, reinforcing these two proteins as interesting components for a vaccine to combat Bcc infections.

## 4. Discussion

A strategy combining bioinformatics and experimental approaches envisaging the identification of surface-exposed and immunogenic proteins of the epidemic strain *B. cenocepacia* J2315 as a representative of Bcc was pursued in this work. The strain was involved in documented patient-to-patient transmission episodes, resulting in several deaths among CF patients [25], and its genome sequence is publicly available [43]. Bcc species are intrinsically resistant to currently available antibiotics including penicillins, narrow-spectrum cephalosporins, aminoglycosides, and polymyxins, and effective strategies for eradicating chronic Bcc infections are lacking [44]. In this context, the development of protective vaccines against Bcc infections becomes highly attractive. To date, no such vaccine is available and only a few studies have been performed envisaging the identification of Bcc immunogenic proteins with the potential to originate a protective and lasting immune response [12,13,14,19,45,46].

Among all the proteins of a pathogen, surface-exposed proteins are the most likely to be involved in the initial steps of pathogen–host interactions and thus to induce immunological responses by the host, being highly attractive for the development of protective vaccines. The bioinformatics survey performed in the present work allowed the identification, within the genome sequence of *B. cenocepacia* J2315, of 263 proteins putatively surface-exposed (including outer membrane and extracellular proteins), 143 of them predicted to contain B-cell epitopes. In order to validate our data, an experimental approach was performed based on the “shaving” approach developed by Rodríguez-Ortega [27], followed by the LC-MS/MS identification of trypsin-digested peptides. To the best of our knowledge, this is the first report of the application of the “surface shaving” technique to identify surface-exposed proteins using live cells of a Bcc member.

Although a total of 263 proteins were predicted as surface-exposed proteins and 143 were bioinformatically predicted as potentially immunogenic, the number of surface-exposed proteins predicted as immunogenic identified by the “shaving” approach was only 16. These differences reinforce the importance of using combined bioinformatics and experimental approaches, since proteins encoded in genomes are not necessarily expressed under specific conditions. Furthermore, genomes often contain incorrect annotations and bioinformatics predictions often fail [47].

In order to identify proteins that were specifically associated with the infection environment, *B. cenocepacia* J2315 cells were cultivated in ASM medium, to mimic the conditions found in the CF lungs. The importance of using culture media mimicking the conditions of infection was highlighted by Liu et al., who found a total of 121 outer membrane proteins produced by *B. cenocepacia* strain Y10 after cultivation in different media, 37 proteins being uniquely identified when the cells were cultivated in ASM [48]. Wolden and colleagues also performed the cell-shaving technique with *Staphylococcus haemolyticus* after the bacterium’s cultivation in the presence of keratinocytes, highlighting the importance of the use of the technique after host–microbe interaction [49].

In addition to the expected surface-exposed proteins, we have also identified cytoplasmic proteins, most probably resulting from cell lysis, estimated to have occurred in about 20% of the cells in the optimized buffer. Variable and, in some cases, much more extensive lysis have also been reported for various Gram-positive bacteria, expected to be more resistant to mechanical stress [17,50]. In the case of Gram-negative bacteria, the amount of predicted cytoplasmic proteins identified using the surface-shaving approach can range from 10 to 62% [17]. Although some fraction of these predicted cytoplasmic proteins can be the result of cell lysis, they can be also proteins that reach the bacterial surface by non-canonical pathways or can be cytoplasmic proteins that are released by the shedding of membrane-vesicle structures containing these proteins [17].

In fact, a fraction of around 63% of the predicted intracellular identified proteins (164 cytoplasmic proteins; seven from the cytoplasmic membrane and four from the periplasm) includes proteins described as moonlighting due to their localization-dependent functions in the cell in other Gram-negative bacteria [39]. Moonlighting proteins can have different functions in different cellular locations, playing important roles in bacterial virulence and host–pathogen interactions [51]. We also do not exclude that some cytoplasmic proteins identified might result from their secretion as part of the cargo of outer-membrane vesicles formed by *B. cenocepacia* J2315.

Regardless of some contamination with cytoplasmic proteins, the shaving approach was successfully used, allowing the identification of proteins predicted as immunogenic, and for three of them, we have confirmed their reactivity with sera from two CF patients with records of infection by Bcc [19]. While the BCAL2958 protein was already cloned and available in our laboratory [26], the cloning, overexpression and purification of the BCAL2645 and BCAL2022 proteins is here reported for the first time. The *B. cenocepacia* proteins BCAL2958 and BCAL2645 are members of the OmpA-like family of proteins, demonstrated for other pathogens as abundant in outer membranes, highly immunogenic and with the capability of eliciting a strong host immune response [52,53]. OmpA-like proteins have also been found to be important in maintaining the integrity of the outer membrane and to be involved in pathogenesis [52,53]. The protein BCAL2958 was previously characterized by our research group as an immunogenic protein and was used in this work as a positive control [26]. More recently, the PspA-like protein BCAL2022 was also identified by immunoproteomics in a study from our research group [19]. PspA-like proteins were previously shown to play a role in the stress response, being important for the survival and virulence of several bacterial pathogens [54]. In *B. pseudomallei*, PspA was found to be important for intracellular survival in a macrophage cell line [55]. Although the serum samples used are quite limited due to the difficulties in gathering samples from CF patients infected with Bcc, the positive reactivity with the selected proteins is here demonstrated. This work is expected to pave the way for future studies including larger CF serum samples from patients infected with different Bcc species, in order to evaluate the universality of these proteins, as well as of others, to be identified as immunogenic proteins, towards the development of a Bcc protective vaccine. Bcc share a relatively high degree of conservation with other pathogenic *B.* species such as *B. mallei* and *B. pseudomallei*. Therefore, the exploitation of common immunogenic proteins would be useful for the development of a broad-spectrum vaccine towards pathogenic members of the *B.* genus.

## 5. Conclusions

The surface-protein identification of live Bcc bacteria, after cell growth in conditions that mimic the CF lung, through a “shaving” approach and further bioinformatics analysis of the immunogenicity of the surface-exposed identified peptides, was for the first time demonstrated to be a reliable method for selecting vaccine candidates or even for the development of new immunodiagnostic tools. The present work was the first step in the optimization of the surface-shaving methodology for Bcc bacteria. In this work, we were able to identify 333 surface-associated proteins, 16 of them being predicted by PSORTb as surface-exposed proteins, 39 as unknown and 175 proteins with intracellular predicted location but previously described as moonlighting proteins in other bacterial species. Three of these proteins, BCAL2958, BCAL2645 and BCAL2022, were validated by Western blotting as immunogenic using serum samples from Bcc-infected CF patients. However, to develop a vaccine able to protect against a broader range of strains/species of Bcc bacteria, the immunogenic and surface-exposed proteins that are conserved and more expressed under conditions that mimic the CF lung have to be identified. Therefore, the surfomes of strains from different Bcc species should be compared for the selection of a panel of vaccine candidates with broader characteristics. The role of the surface-exposed proteins here identified and their exposed peptides in the interaction with the host will be also interesting to study.

## Figures and Tables

**Figure 1 vaccines-08-00509-f001:**
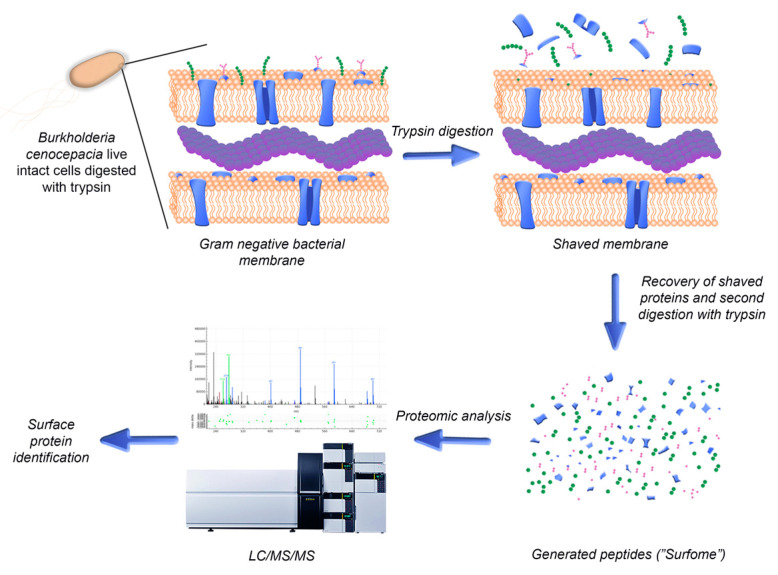
Schematic representation of the steps involved in the identification of surface-exposed proteins using the surface-shaving approach for live *B. cenocepacia* cells.

**Figure 2 vaccines-08-00509-f002:**
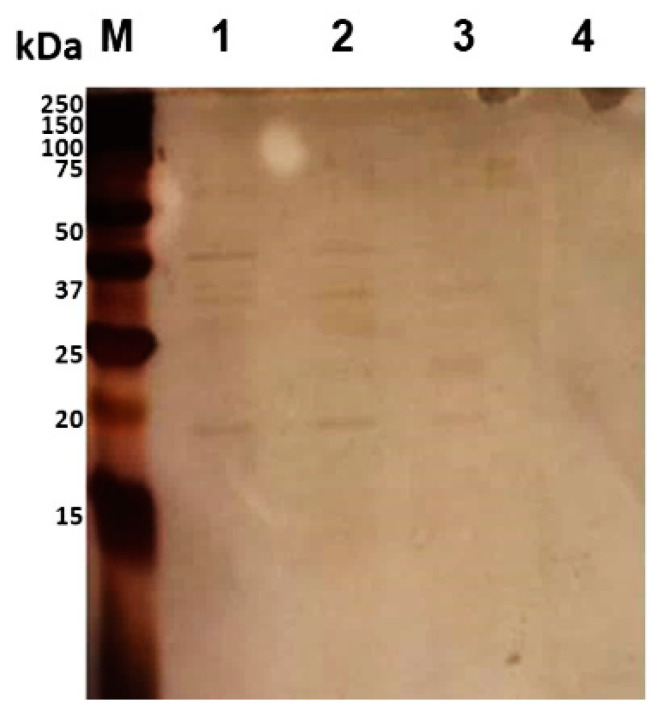
Separation of trypsin-digested mixtures of *B. cenocepacia* J2315 by 15% SDS-PAGE, followed by silver staining. Lanes: **M**—Precision Plus Protein^TM^ DualXtra Standard (BIORAD); **1**—bacteria shaved with trypsin (5 μg/mL) in PBS, pH 7.4, for 30 min at 37 °C; **2**—bacteria shaved with trypsin (5 μg/mL) in PBS for 10 min at 37 °C; **3**—bacteria shaved with trypsin (2.5 μg/mL) in PBS for 10 min at 37 °C; **4**—control sample—bacteria without trypsin digestion in PBS.

**Figure 3 vaccines-08-00509-f003:**
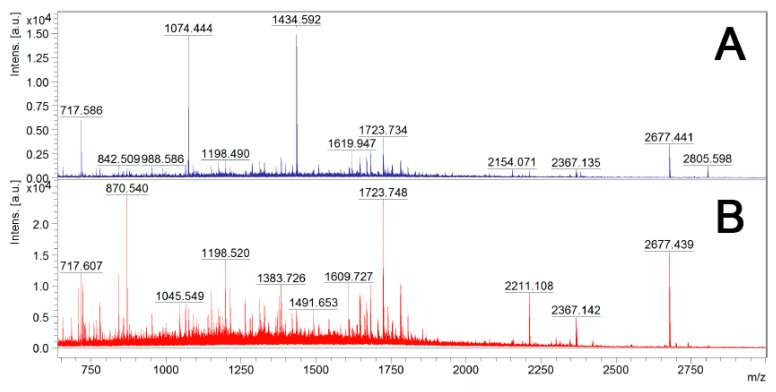
Mass spectra from MALDI-TOF MS of trypsin-digested samples of live *B. cenocepacia* J2315 cells. **Sample A**—Single trypsin digestion: 2.5 μg/mL of trypsin in PBS for 10 min at 37 °C; **Sample B**—Two trypsin digestions: 2.5 μg/mL of trypsin in PBS for 10 min at 37 °C, and an additional trypsin (1 μg/mL) digestion overnight at 37 *°*C, after cell debris removal by centrifugation and filtration.

**Figure 4 vaccines-08-00509-f004:**
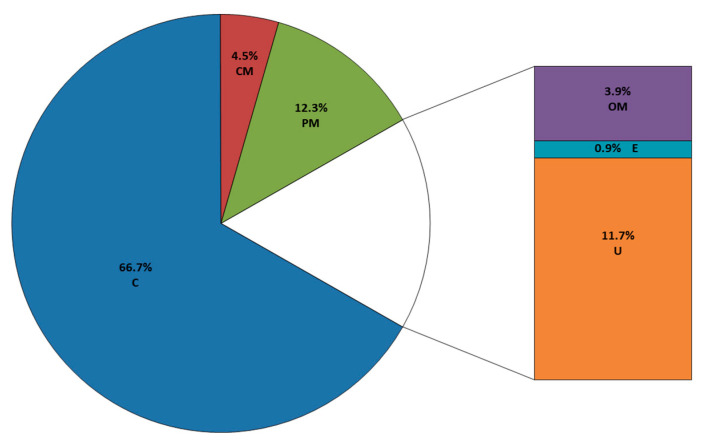
Percentage representation of the subcellular localization of the surface-shaving-identified proteins from *B. cenocepacia* J2315, as determined by PSORTb 3.0.2. The 333 proteins (≥5 peptides within a single protein) were classified in the following categories: outer membrane (**OM**), extracellular (**E**), periplasm (**PM**), cytoplasmic membrane (**CM**), cytoplasm (**C**) and unknown (**U**).

**Figure 5 vaccines-08-00509-f005:**
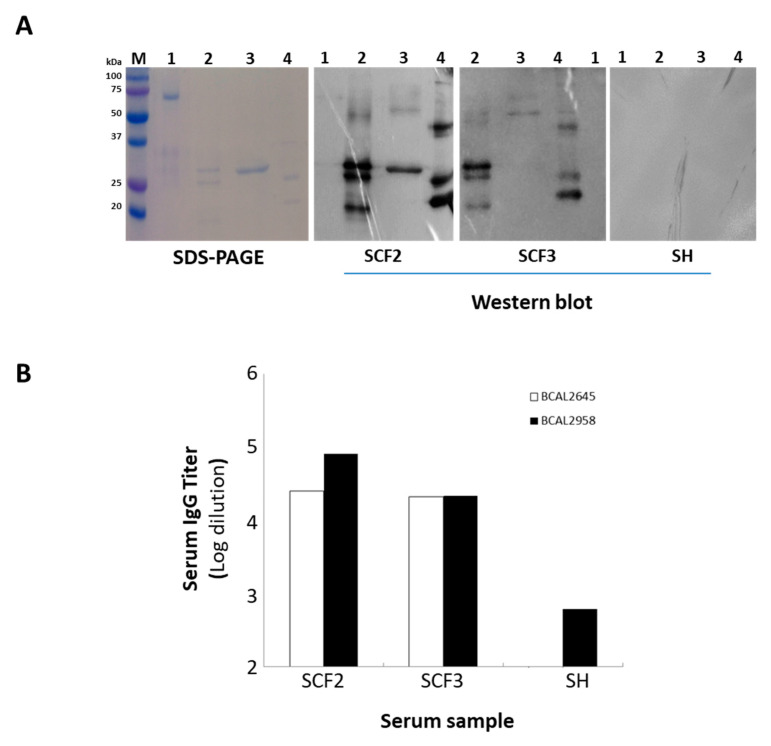
**A**—SDS-PAGE and Western blot of the purified recombinant proteins BCAL2022, BCAL2645 and BCAL2958 from *B. cenocepacia* J2315 probed with the human serum samples SCF2 and SCF3 from CF patients with positive cultures of Bcc bacteria or a pool of human serum samples from healthy donors (SH). Lanes: M—Precision Plus ProteinTM DualXtra Standard (BIORAD); 1—BSA; 2—purified recombinant protein BCAL2958; 3—purified recombinant protein BCAL2022; 4—purified recombinant protein BCAL2645; **B**—levels of IgG antibody against BCAL2645 and BCAL2958 proteins present in serum samples from CF patients with clinical history of Bcc infection (SCF2 and 3) and in healthy individuals (SH). Serum antibody concentrations were defined as endpoint titers and were calculated as the reciprocal of the highest serum dilution producing an OD_450_ above the cut-off value. The cut-off value was determined as the mean OD_450_ negative control plus 3 standard deviations. Titers of ≥700 and ≥100 were considered as positive for BCAL2958 and BCAL2645, respectively. The IgG titer data for BCAL2958 were previously obtained in [26].

**Table 1 vaccines-08-00509-t001:** Bacterial strains and plasmids used in this work. Ap^R^, resistant to ampicillin; Kan^R^, resistant to kanamycin.

Strain or Plasmid	Genotype or Description	Source or Reference
*B. cenocepacia* J2315	Cystic Fibrosis isolate; ET12 lineage reference strain, genome sequence available.	[25]
*E. coli* DH5α	F^−^ Φ80*lac*ZΔM15 Δ(*lac*ZYA-*arg*F) U169 *rec*A1 *end*A1 *hsd*R17(r_k_^−^, m_k_^+^) *pho*A *sup*E44 *thi*-1 *gyr*A96 *rel*A1 λ^−^	Invitrogen (Carlsbad, CA, USA)
*E. coli* BL21 (DE3)	F^−^ *ompT hsdSB* (rB−mB−) dcm *gal λ*(DE3)	Stratagene (San Diego, CA, USA)
**Plasmids**		
pET23a+	Cloning/expression vector, T7 promoter, C-terminal 6x His-Tag, Ap^r^	Novagen (Madison, WI, USA)
pET29a+	Cloning/expression vector, T7 promoter, thrombin recognition site, C-terminal 6’ His-Tag, Kan^r^	Novagen
pSAS38	pET23a+ containing the pET29a+ thrombin recognition site, cloned downstream of the T7 promoter and upstream of the C-terminal 6× His-Tag, Ap^r^	This study
pMM1	pSAS38 containing the *BCAL2022* gene, cloned downstream of the T7 promoter, Ap^r^	This study
pSAS36	pET23a+ containing the *BCAL2645* gene, cloned downstream of the T7 promoter and upstream of the C-terminal 6× His-Tag, Ap^r^	This study
pSAS6	pET23a+ containing the *BCAL2958* gene, cloned downstream of the T7 promoter and upstream of the C-terminal 6× His-Tag, Ap^r^	[26]

**Table 2 vaccines-08-00509-t002:** Summary of surface-associated proteins identified in *B. cenocepacia* J2315 by using the “surface shaving” strategy followed by LC-MS/MS (threshold score: ≥5 peptides). Their subcellular localization, associated biological functions, numbers of unique peptides identified and probabilities of containing predicted immunogenic epitopes are shown.

ORF ^1^	Description ^1^	Peptides Identified (PSMs) ^2^	Subcellular Localization ^3^	Domains ^4^	B-Cell Epitope Average (Predicted Peptides) ^5^
BCAL1416 (AlaS)	Alanyl-tRNA synthetase	23 (28)	U (Class 3)	tRNA synthetases class II (A) (PF01411) Threonyl and Alanyl tRNA synthetase second additional domain (PF07973) DHHA1 domain (PF02272)	0.464 (23)
BCAM0965 (mdh)	Malate dehydrogenase	17 (29)	U (Class 3)	Lactate/malate dehydrogenase, NAD binding domain (PF00056) Lactate/malate dehydrogenase, alpha/beta C-terminal domain (PF02866)	0.430 (6)
BCAL2736	Isocitrate dehydrogenase	17 (25)	U (Class 3)	Isocitrate/isopropylmalate dehydrogenase (PF00180)	0.460 (8)
BCAL3203 (TolB)	Translocation protein TolB	17 (22)	U (Class 3)	TolB amino-terminal domain (PF04052) WD40-like Beta Propeller Repeat (PF07676)	0.454 (9)
BCAL2777	Putative N-acetylmuramoyl-L-alanine amidase	14 (15)	U (Class 3)	AMIN domain (PF11741) N-acetylmuramoyl-L-alanine amidase (PF01520)	0.511 (12)
BCAL2956	Hypothetical protein	13 (32)	U (Class 3)	DUF2059 (PF09832)	0.485 (6)
BCAL2993 (PepN)	Aminopeptidase N	13 (14)	U (Class 3)	Peptidase M1 N-terminal domain (PF17900) Peptidase family M1 domain (PF01433) DUF3458, Ig-like fold (PF11940) DUF3458_C, ARM repeats (PF17432)	0.456 (24)
BCAL3311 (BcnA)	Secreted bacterial lipocalin	12 (22)	E (Class 1)	YceI-like domain (PF04264)	0.455 (3)
BCAL1262 (CarB)	Carbamoyl phosphate synthase large subunit	12 (13)	U (Class 3)	Carbamoyl-phosphate synthase L chain: ATP binding domain (PF02786); Oligomerization domain (PF02787) MGS-like domain (PF02142)	0.457 (25)
BCAM1931	Putative porin	11 (15)	OM (Class 3)	Porin_4 (PF13609)	0.503 (10)
BCAL0765	Hypothetical protein	11 (12)	U (Class 3)	ABC transporter substrate-binding protein (PF04392)	0.454 (5)
BCAL1961	Hypothetical protein	10 (20)	U (Class 3)	Ankyrin repeats (PF12796)	0.472 (6)
BCAM0906	Putative dienelactone hydrolase family protein	10 (14)	U (Class 3)	Dienelactone hydrolase family (PF01738)	0.471 (7)
BCAL2206 (PhaP)	Phasin-like protein	10 (13)	U (Class 3)	Phasin_2 protein (PF09361)	0.482 (5)
BCAM0043	Hypothetical protein	10 (12)	U (Class 3)	Phage late control gene D protein (PF05954) T6SS_Vgr (PF13296) DUF2345 (PF10106)	0.494 (12)
BCAM1576	Phosphoesterase family protein	10 (10)	E (Class 3)	Phosphoesterase family (PF04185)	0.489 (10)
BCAL0340 (BscM)	Putative lipoprotein. Part of the T6SS gene cluster	9 (13)	U (Class 3)	Tetratricopeptide repeat (PF14559)	0.506 (6)
BCAL1493	Putative exported protein	9 (12)	U (Class 3)	NlpB/DapX lipoprotein (PF06804)	0.518 (6)
BCAL2166	Putative lipoprotein	9 (10)	U (Class 3)	NlpB/DapX lipoprotein (PF06804)	0.500 (10)
BCAL0305	Hypothetical protein	8 (21)	U (Class 3)	MlaC protein (PF05494)	0.480 (7)
BCAL1893	Family M23 peptidase	8 (13)	OM (Class 3)	LysM domain (PF014769) Peptidase family M23 (PF01551)	0.523 (4)
BCAL0151	Putative lipoprotein	8 (12)	U (Class 3)	Periplasmic binding protein (PF13458)	0.439 (10)
BCAL2934 (EtfA)	Electron transfer flavoprotein alpha-subunit	8 (10)	U (Class 3)	Electron transfer flavoprotein domain (PF01012) Electron transfer flavoprotein FAD-binding domain (PF00766)	0.443 (3)
BCAM1012	Putative histone-like protein	8 (10)	U (Class 3)	Bacterial DNA-binding protein (PF00216)	0.520 (2)
BCAL2958	Putative OmpA family protein	7 (25)	OM (Class 3)	OmpA family (PF00691)	0.512 (3)
BCAM2827	Hypothetical protein	7 (13)	U (Class 3)	MlaC protein (PF05494)	0.462 (4)
BCAS0667	Uncharacterized protein	7 (9)	U (Class 3)	Phage late control gene D protein (PF05954)	0.500 (20)
BCAL0020	Putative branched-chain amino acid ABC transporter periplasmic substrate binding protein	7 (9)	U (Class 3)	Periplasmic binding protein (PF13458)	0.442 (8)
BCAS0219	Hypothetical protein	7 (8)	U (Class 3)	Outer membrane lipoprotein-sorting protein (PF17131)	0.461 (7)
BCAM2736	Hypothetical protein	7 (8)	U (Class 3)	none	0.534 (0)
BCAM1712 (PaaH)	3-hydroxy-acyl-CoA dehydrogenase	7 (8)	U (Class 3)	3-hydroxyacyl-CoA dehydrogenase: NAD binding domain (PF02737); C-terminal domain (PF00725) 3-hydroxybutyryl-CoA dehydrogenase reduced Rossmann-fold domain (PF18321)	0.443 (7)
BCAL2687	Putative lipoprotein	7 (7)	U (Class 3)	Periplasmic binding protein (PF13458)	0.438 (7)
BCAL2946 (udg)	UDP-glucose dehydrogenase	7 (7)	U (Class 3)	UDP-glucose/GDP-mannose dehydrogenase family: NAD binding domain (PF03721); Central domain (PF00984); UDP binding domain (PF03720)	0.447 (11)
BCAL2828	Hypothetical protein	6 (11)	U (Class 3)	Carboxypeptidase regulatory-like domain (PF13620)	0.468 (3)
BCAL3204 (Pal)	Peptidoglycan-associated lipoprotein	6 (9)	OM (Class 3)	OmpA family (PF00691)	0.507 (3)
BCAS0147	Hypothetical protein	6 (7)	U (Class 3)	Lactonase, 7-bladed beta-propeller (PF10282)	0.463 (12)
BCAL3228	Hypothetical protein	6 (7)	U (Class 3)	none	0.516 (3)
BCAL2022	PspA/IM30 family protein	6 (6)	OM (Class 3)	PspA/IM30 family (PF04012)	0.503 (3)
BCAL1985	Putative peptidylprolyl isomerase	6 (6)	OM (Class 3)	PPIC-type PPIASE domain (PF13616)	0.494 (5)
BCAL2083 (YaeT)	Outer membrane protein assembly factor YaeT	6 (6)	OM (Class 3)	Surface antigen variable number repeat POTRA (PF07244) Omp85 (PF01103)	0.488 (19)
BCAM0900	Hypothetical protein	6 (6)	U (Class 3)	Peptidase propeptide and YPEB domain (PF13670)	0.537 (0)
BCAL0377	Subfamily M24B metalopeptidase	6 (6)	U (Class 3)	Creatinase/prolidase N-terminal domain (PF01321) (PF16189) Metallopeptidase family M24 (PF00557) C-terminal region of peptidase_M24 (PF16188)	0.455 (12)
BCAM1833 (AcnB)	Bifunctional aconitate hydratase 2/2-methylisocitrate dehydratase	6 (6)	U (Class 3)	Aconitate B, N-terminal domain (PF11791) Aconitate hydratase 2, N-terminus (PF06434) Aconitase family (PF00330)	0.469 (16)
BCAL0804	Hypothetical protein	6 (6)	U (Class 3)	Tetratricopeptide repeat (PF14559) Tetratricopeptide repeat (PF13432)	0.455 (20)
BCAM2761 (CblA)	Giant cable pilus	5 (12)	E (Class 3)	CS1 type fimbrial major subunit (PF04449)	0.501 (2)
BCAL2645	Putative OmpA family protein	5 (11)	OM (Class 3)	YMGG-like Gly-zipper (PF13441) OmpA family (PF00691)	0.502 (4)
BCAL1288	Family M23 peptidase	5 (9)	OM (Class 3)	LysM domain (PF01476) Peptidase family M23 (PF01551)	0.513 (5)
BCAL2738	Hypothetical protein	5 (7)	U (Class 3)	DUF192 (PF02643)	0.461 (4)
BCAL0389 (DsbC)	Thiol:disulfide interchange protein	5 (7)	U (Class 3)	Disulfide bond isomerase protein N-terminus (PF10411) Thioredoxin-like domain (PF13098)	0.481 (3)
BCAL3035 (TrxB)	Thioredoxin reductase	5 (6)	U (Class 3)	Pyridine nucleotide-disulphide oxidoreductase (PF07992)	0.449 (6)
BCAL1881 (BamB)	Outer membrane protein assembly factor BamB	5 (5)	OM (Class 3)	PQQ-like domain (PF13360)	0.444 (8)
BCAM2549 (OpcM)	Multidrug efflux system outer membrane protein	5 (5)	OM (Class 1)	Outer membrane efflux protein (PF02321)	0.521 (11)
BCAL0304	VacJ-like lipoprotein	5 (5)	OM (Class 3)	MlaA lipoprotein (PF04333)	0.469 (4)
BCAL0342 (BcsK)	Putative type VI secretion system protein TssC	5 (5)	OM (Class 3)	EvpB/VC_A0108, tail sheath N-terminal domain (PF05943)	0.472 (15)
BCAL0273 (CyaY)	Frataxin-like protein	5 (5)	U (Class 3)	Frataxin-like domain (PF01491)	0.465 (2)

^1^ Burkholderia Genome Database [30,40]. ^2^ Number of unique peptides identified using surface shaving with trypsin and LC-MS/MS analysis. PSMs-peptide-spectrum matches. ^3^ Confidence level: Class 1-Subcellular localization experimentally demonstrated in the same species; Class 2-Subcellular localization of highly similar gene experimentally demonstrated in another organism OR a paralog experimentally demonstrated in the same organism. BLAST Expect value of 10e^−10^ for query within 80–120% of subject length; Class 3-Subcellular localization computationally predicted by PSORTb V3.0 [32]. Abbreviations: OM-outer membrane; E-extracellular; U-unknown. ^4^ Pfam Database [31,41]. ^5.^ BepiPred-2.0 server [35,42]. Threshold of 0.5 used. Peptides shorter than 5 or larger than 25 amino acids were not considered.

**Table 3 vaccines-08-00509-t003:** Selected surface-associated proteins identified in *B. cenocepacia* J2315 by using the “surface shaving” strategy followed by LC-MS/MS (threshold score: ≥5 peptides) and their predicted B-cell epitopes. The peptides identified by the surface-shaving approach and predicted as B-cell epitopes are shown in bold.

ORF	Description	Peptides Identified ^1^	Predicted B-Cell Epitopes ^2^
BCAL2958	OmpA family protein	^93^ITYQADALFDFDK^105^ ^93^ITYQADALFDFDKATLKPLGK^113^ ^114^QKLDELASK^122^ ^116^LDELASK^122^ ^123^IEGMNTEVVVATGYTDR^139^ ^203^RVEVEVVGTQQVQK^216^ ^204^VEVEVVGTQQVQK^216^	^101^**FDFDKATLKPLGKQ**^114^ ^139^**R**IGSDKYNDRL^149^ ^211^**TQQVQK**TTV^219^
BCAL2645	OmpA family protein	^79^LAPSAAQTGTQVTEQPDGSLK^99^ ^161^AQSVVNALVQR^171^ ^178^LSAQGMGASNPIADNATEAGR^198^ ^203^RVEIYLR^209^ ^204^VEIYLR^209^	^88^**TQVTEQ**^93^ ^109^ATNQYAITPA^118^ ^145^DSTGSAQLNQTL^156^ ^183^**MGASNPIADNATEAGR**AQN^201^
BCAL2022	PspA/IM30 family protein	^13^GLLNDAADSVQDPSR^27^ ^35^ELDDSIGR^42^ ^43^AENSLIEIEAQVATQR^58^ ^78^ALQGGDEALAR^88^ ^89^EALAAQSNAEAER^101^ ^150^DVAASALGGIGGK^162^	^20^**DSVQDPSRD**^28^ ^156^**LGGIGGK**NLSEDFQKLEDK^174^ ^215^AALKKQLD^222^

^1^ Peptides identified using surface-shaving with trypsin and LC-MS/MS analysis. ^2^ BepiPred-2.0 server [35,42]. Threshold of 0.5 used. Peptides shorter than 5 or larger than 25 amino acids were not considered.

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
