# Peer review of "Characterization of the Burkholderia cenocepacia J2315 Surface-Exposed Immunoproteome"

_vaccines, 2020, doi:10.3390/vaccines8030509_

Round 1
Reviewer 1 Report
Excellent manuscript, a very enjoyable read, well done!
Just one suggestion: the authors should comment/speculate, why purified recombinant proteins BCAL2958 and BCAL2645 form several bands in SDS-PAGE and the immuno-blots shown in Fig.5. What is the theoretical molecular weight of these proteins? Multimers? Processing? Degradation?
Finally, a typo on page 2, line 77: ‘itis’, blank missing
Author Response
Dear Reviewer,
We are very happy for your kind comments and thankful for the suggestions.
Below follows your suggestions and the actions taken:
Reviewer comment:Â the authors should comment/speculate, why purified recombinant proteins BCAL2958 and BCAL2645 form several bands in SDS-PAGE and the immuno-blots shown in Fig.5. What is the theoretical molecular weight of these proteins? Multimers? Processing? Degradation?
Thanks for the suggestion. We have included the following explanation, new lines 438-446:
Analysis of the BCAL2958, BCAL2645 and BCAL2022 amino acid sequences using ProtParam program revealed that these proteins have a predicted molecular weight of 23.9 kDa, 21.6 kDa and 24.3 kDa, respectively. After SDS-PAGE, at least 3 different molecular weight forms of the protein BCAL2958 and two for BCAL2645 (Fig. 5a). The multiple bands most probably result from the different stages of processing and translocation of the proteins from the cytoplasm to the outer membrane, as previously reported for BCAL2958 protein and for other OmpA-like proteins [26]. In the case of BCAL2958, the first two forms with higher molecular mass were described to correspond to the native pre-protein with a 6× His-tag containing the signal peptide, and the mature protein with a 6× His-tag without the signal peptide, respectively [26].
Reviewer: Finally, a typo on page 2, line 77: ‘itis’, blank missing
Answer: the typo was fixed.
Kind regards
Jorge Leitão (on behalf of all co-authors)
Â
Â
Â
Reviewer 2 Report
Major concerns:
1. The authors checked immunogenicity of 3 proteins from a surface exposed protein pool. They conducted the experiments by immunoblotting. To obtain more informative data, an ELISA assay could be performed.
2. Some typographical errors need to be corrected. For the example, Burkholderia cepacia should be italic, and abbreviation of the organism should be used when it appears in the second time and thereafter through the text.Â
Author Response
Dear Reviewer,
Thanks for your comments and concerns.Â
Below follows our answers to your concerns:
Major concerns:
- The authors checked immunogenicity of 3 proteins from a surface exposed protein pool. They conducted the experiments by immunoblotting. To obtain more informative data, an ELISA assay could be performed.
ANSWER: Thanks for the suggestion. We have performed ELISA assays for two proteins, as BCA2022 only showed faint bands with one serum.Â
A new part was added to the materials and methods, new lines 252-274, Figure 5 now has a new panel B, with ELISA results, and an extended legend, new lines 430-436, and a new sentence was added to the Results section, new lines 446-448
- new lines 252-274
"2.12. Enzyme‑linked immunosorbent assay (ELISA)
Quantification of the IgG levels present in sera samples from CF patients with clinical history of Bcc infection against purified 6× His-tagged BCAL2645 were determined by enzyme-linked immunosorbent assay (ELISA) as previously described [26]. Briefly, the BCAL2645 protein was diluted to 2 μg/ml in 100 mM sodium carbonate buffer (pH 9.6), and 100 μl was applied per well to 96-wells ELISA plates (Greiner Microlon 600, Greiner Bio-One). After overnight incubation at 4 °C, the plates were blocked with 250 μl of 3 % BSA/PBS overnight at 4 °C. Serum samples to be tested were serially diluted (1:100 to 1:10,000) in 3 % BSA / PBS-T (PBS supplemented with 0.05 % Tween 20). After addition of the diluted serum samples to the plates, they were incubated 2 hours at 25 °C. Then, the plates were washed with PBS-T and 100 μl of HRP-conjugated rabbit anti-Human IgG (SANTA CRUZ Biotechnology) antibody at 1:3000 dilution in 3 % BSA / PBS-T was added to the plates. After incubation at 25 °C for 1 hour, the plates were washed with PBS-T. Then, 100 μl of the peroxidase substrate 3,3′,5,5′-tetramethylbenzidine (TMB, SIGMA) were added. After incubation for 20 minutes at 25 °C, the reaction was stopped by the addition of 100 μl of 0.5 M H2SO4. The plates were read at 450 nm in the SPECTROstar Nano microplate reader (BMG LABTECH). A pool of sera from healthy humans (Sigma) was used as control. Internal positive and negative controls were included in each plate. All serum samples were analyzed in triplicate using at least two independent experiments.
Serum antibody concentrations were defined as endpoint titers and were calculated as the reciprocal of the highest serum dilution producing an OD450 above the cut-off value. The cut-off value was determined as the mean OD450 of the negative control plus 3 standard deviations. A titer above the cut-off value was considered positive for the ELISA."
- new lines 430-436 (legend of new Fig. 5):
"B - IgG antibody levels against BCAL2645 and BCAL2958 proteins present on sera samples from CF patients with clinical history of Bcc infection (SCF2 and 3) and in healthy individuals (SH). Serum antibody concentrations were defined as endpoint titers and were calculated as the reciprocal of the highest serum dilution producing an OD450 above cut-off value. The cut-off value was determined as the mean OD450 negative control plus 3 standard deviations. A titer of ≥700 and ≥ 100 was considered as positive for BCAL2958 and BCAL2645, respectively. The IgG titers data for BCAL2958 were previously obtained in [26]."
- new lines 446-448:
"The serum samples from CF patients with a Bcc infection record presented high antibody titers against BCAL2645 and BCAL2958, reinforcing these two proteins as interesting components of a vaccine to combat Bcc infections."
Â
2. Some typographical errors need to be corrected. For the example, Burkholderia cepacia should be italic, and abbreviation of the organism should be used when it appears in the second time and thereafter through the text.Â
ANSWER: Thanks for the indication. We have gone through the text and corrected the typos indentified. The name of th eorganism is now in italics, expept when the meaning is the Genome Database, so we kept Burkholderia Genome Database, which has no taxonomical meaning.
Â
Round 2
Reviewer 2 Report
I do not have further concerns.